# *De novo* phosphoinositide synthesis in zebrafish is required for triad formation but not essential for myogenesis

**Lindsay Smith[1,2], Lacramioara Fabian[1], Almundher Al-Maawali[1,3], Ramil R. Noche[4], James J. Dowling**[1,5,6]*

**1** Genetics and Genome Biology Program, The Hospital for Sick Children, Toronto, Ontario, Canada, **2** Ontario Institute for Cancer Research, Toronto, Ontario, Canada, **3** Department of Genetics, College of Medicine and Health Sciences, Sultan Qaboos University & Sultan Qaboos University Hospital, Muscat, Oman, **4** Zebrafish Genetics and Disease Models Core Facility, The Hospital for Sick Children, Toronto, Ontario, Canada, **5** Division of Neurology, The Hospital for Sick Children, Toronto, Ontario, Canada, **6** Departments of Paediatrics and Molecular Genetics, University of Toronto, Toronto, Ontario, Canada

* james.dowling@sickkids.ca

**Data Availability Statement:** The majority of relevant data are contained within the manuscript and/or supporting information. The data underlying the results presented in the study are available on

## Abstract

Phosphoinositides (PIPs) and their regulatory enzymes are key players in many cellular processes and are required for aspects of vertebrate development. Dysregulated PIP metabolism has been implicated in several human diseases, including a subset of skeletal myopathies that feature structural defects in the triad. The role of PIPs in skeletal muscle formation, and particularly triad biogenesis, has yet to be determined. CDP-diacylglycerol-inositol 3-phosphatidyltransferase (CDIPT) catalyzes the formation of phosphatidylinositol, which is the base of all PIP species. Loss of CDIPT should, in theory, result in the failure to produce PIPs, and thus provide a strategy for establishing the requirement for PIPs during embryogenesis. In this study, we generated *cdipt* mutant zebrafish and determined the impact on skeletal myogenesis. Analysis of *cdipt* mutant muscle revealed no apparent global effect on early muscle development. However, small but significant defects were observed in triad size, with T-tubule area, inter terminal cisternae distance and gap width being smaller in *cdipt* mutants. This was associated with a decrease in motor performance. Overall, these data suggest that myogenesis in zebrafish does not require *de novo* PIP synthesis but does implicate a role for CDIPT in triad formation.

## Introduction

The primary function of skeletal muscle is to produce the force that initiates and controls movement. Muscle has a number of unique substructures that are dedicated to force production, including the sarcomere, the neuromuscular junction (NMJ) and the triad [1]. As our understanding of the molecular basis of human muscle diseases grows, it is becoming more apparent that many myopathies involve alterations to at least one of these structures [1–3]. Of increasing significance are the abnormalities in the structure and function of the triad, which

figshare. DOI: https://doi.org/10.6084/m9.figshare.12490589.v1.

**Funding:** JJD - # - National Science and Engineering Research Council (NSERC) JJD - # - Canadian Institutes of Health Research (CHIR) 324830 and 376691 The funders had no role in study design, data collection and analysis, decision to publish, or preparation of the manuscript.

**Competing interests:** The authors have declared that no competing interests exist.

represents the apposition of sarcolemmal invaginations called T-tubules and the terminal cisternae of the sarcoplasmic reticulum (SR). The key role of the triad is to mediate excitation-contraction coupling (ECC), the process by which skeletal muscle translates neuronal signals into muscle contraction [4, 5].

Triad malformations are considered the major driver of muscle weakness in many myopathies [1, 6]. For example, loss of function mutations in *CACNA1S* and *RYR1*, genes that encode two critical components of the triad, result in subtypes of congenital myopathies, severe childhood conditions characterized by muscle weakness and severe disability [6, 7]. There are also a group of conditions referred to as secondary triadopathies, where gene mutations result in secondary structural and/or functional disruption of the triad [1]. Most relevant for this study are the centronuclear myopathies (discussed further below), the genetic causes of which primarily encode proteins that regulate membrane trafficking [5, 8, 9].

The factors that govern the development and maintenance of the triad remain unclear. Recent data has suggested that phosphoinositides may play an important role in triad formation and/or maintenance. Phosphoinositides (PIPs) are a family of membrane phospholipids involved in many essential cell functions, including cellular signaling, endocytosis, and autophagy, and are present in almost all cell types across eukaryotic species [10, 11]. Formation and turnover of the various PIP species are catalyzed by evolutionarily conserved families of kinases and phosphatases [12, 13]. Dysregulation of PIPs and their metabolic enzymes has been implicated in a number of human diseases, such as centronuclear myopathy, Charcot-Marie-Tooth Disease (CMT), Alzheimer's disease, and some forms of cancer [14–17]

The consideration of a potential role for PIPs in muscle development comes from two areas of study. One is the work surrounding BIN1, a BAR domain-containing protein that is known to recognize and induce membrane curvature [18, 19]. BIN1 has a PIP-binding domain that interacts with PIP2 (one of the seven PIP sub-species), and this interaction plays a critical role in the formation of T-tubules [20]. Recessive mutations in *BIN1* result in centronuclear myopathy, a severe congenital muscle disease featuring abnormal muscle structure including disturbance of the T-tubule and the triad as a whole [21]. The second line of evidence comes from another form of centronuclear myopathy called X-linked myotubular myopathy or XLMTM [22]. XLMTM is caused by mutations in the PIP phosphatase myotubularin [23]. Mutation in myotubularin causes accumulation of PI3P and leads to abnormalities in the appearance and number of the triad [24, 25].

In this study, we investigated the role of PIPs in skeletal muscle triad development using the zebrafish model system. Zebrafish is an elegant model for studying skeletal muscle development [26–31]. Skeletal muscle develops rapidly in zebrafish, muscle fibers are already developing by 24 hours post fertilization (hpf), with elongated fibers visible by 2 days post fertilization (dpf) [32]. Skeletal muscle is highly prominent in embryos and larvae, and the transparency of developing fish allows muscle fibers to be easily observed [28]. Additionally, zebrafish muscle shares many structural and histological features with mammalian muscle [33].

To determine the overall requirement for PIPs in muscle development we used the CRISPR/Cas9 technology to generate a *cdipt* zebrafish mutant. CDIPT, also known as phosphatidylinositol synthase (PIS), catalyzes the addition of a *myo*-inositol ring to a phospholipid backbone, cytidine diphosphate-diacylglycerol (CDP-DAG), to generate the base of all PIPs, phosphatidylinositol (PI) [34] (Fig 1A). It is the only protein currently known to perform this function in zebrafish [35]. CDIPT is a highly conserved integral membrane protein found on the cytoplasmic side of the endoplasmic reticulum (ER).

Previous study of a zebrafish *cdipt* mutant revealed a liver phenotype reminiscent of phenotypes seen in other models of PIP dysregulation [35]. This study, however, did not examine skeletal muscle. In the current study, we examine the skeletal muscle in a new *cdipt* mutant.

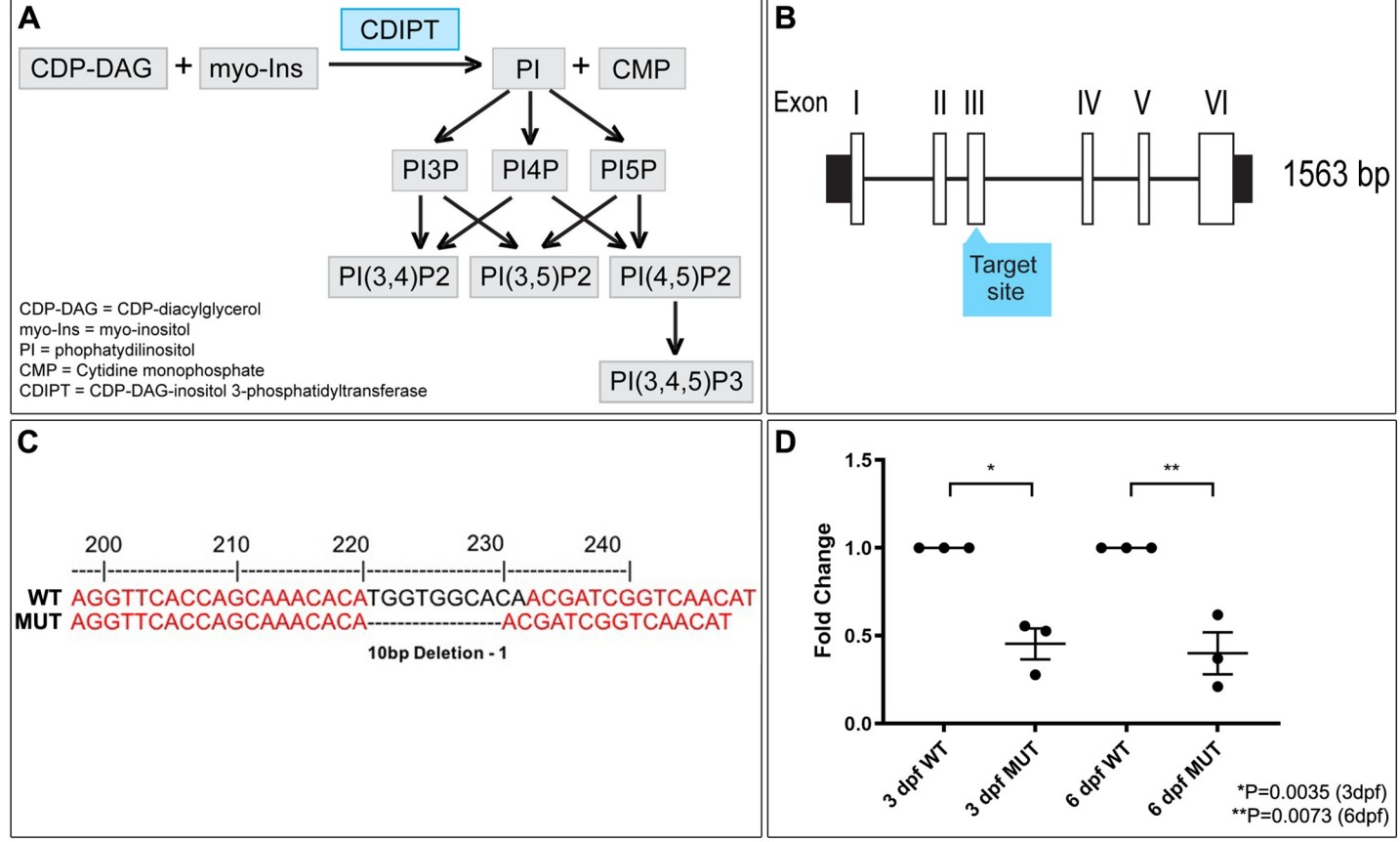

**Fig 1. Development of a CRISPR/Cas9 *cdipt* mutant zebrafish. A)** Schematic representation of phosphoinositide signaling pathway. CDIPT catalyzes the addition of the myo-inositol to the CDP-DAG to generate PI, which is the base precursor for all species of PIPs. **B)** Schematic representing exon organization of *cdipt*. Exon 3 was targeted by CRISPR/Cas9 gene editing. **C)** Sanger sequencing of wildtype (WT) and homozygous *cdipt* mutant (MUT) larvae showing a 10-bp deletion in exon 3 of *cdipt*. **D)** Fold change of mRNA levels between WT and MUT fish at both 3 dpf and 6 dpf. There is a significant change in *cdipt* mRNA levels between WT and MUT zebrafish at both 3 dpf (0.5-fold reduction; *p = 0.0035) and 6 dpf (0.6-fold reduction; **p = 0.0073). Each replicate is represented by a point, n = 30 per replicate; Student's *t* test, 2-tailed. Error bars indicate SEM.

We show that loss of CDIPT has no effect on early muscle development, suggesting that skeletal myogenesis does not require *de novo* PIP synthesis. Instead, CDIPT appears to be required for proper formation of the triad.

## Materials and methods

### Zebrafish maintenance

Zebrafish stocks were maintained at the Zebrafish Facility at the Hospital for Sick Children, Toronto, ON, Canada. All zebrafish procedures were performed in compliance with the Animals for Research Act of Ontario and the Guidelines of the Canadian Council on Animal Care. Approved by Animal Care Committee at SickKids, Toronto, Ontario, Canada. Assigned Protocol #: 1000052731. Zebrafish were anesthetized using Tricaine. The *cdipt*^hi559^ zebrafish line was obtained from the Zebrafish International Resource Center (ZIRC)

### Generation of zebrafish *cdipt* mutants

A detailed procedure for CRISPR/Cas9 editing in zebrafish has been described previously [36]. Blast search of the zebrafish genome confirmed the existence of a single *cdipt* paralog in

zebrafish (NM_207088). The *cdipt* target in this study was 5'- GGTTCACCAGCAAACACA TGGTGG-3' in exon 3. One-cell-stage AB WT embryos were injected with gRNA and Ca9 mRNA with a Picopump (World Precision Instruments). Potential founders ($F_0$) were out-crossed to AB WT fish. Genomic DNA was isolated from single $F_1$ embryos at 6 dpf and geno-typed using high resolution melt (HRM) analysis. A *cdipt* sequence spanning the CRISPR/Cas9 target site was amplified with the following primers: F: 5'-AGCTGGAACAGAAAAGTG TAGGA-3'; and R: 5'-TAGGTACAAAATTTGGTGCAATG-3'. Carriers were identified and outcrossed ultimately to the $F_3$ generation. In-cross progeny from the $F_3$ and F4 generations were characterized in this study.

## Real-time PCR (qPCR)

RNA was extracted from 3 dpf and 6 dpf *cdipt* mutant zebrafish and their wildtype siblings using RNAeasy (Qiagen). RNA samples were reverse transcribed into cDNA using the iScript cDNA synthesis kit (BioRad). Primers were designed to result in a product spanning exons 4–6 of *cdipt*: F: 5'-ACCCCATTTTACGGCTGTACT-'3; and R: 5'-TACCTGGGGTTCTTC GATGT-'3. Products were amplified using Step-One-Plus Real-Time PCR System (Applied Biosystems). The zebrafish beta-actin gene, *actb1*, was used as an endogenous control.

## Birefringence

Tricaine-anaesthetised larvae were mounted in 3% methylcellulose on glass slides and imaged under polarized light on a dissecting microscope (Olympus SZX7).

## Skeletal myofiber preparations

Myofiber preparations of 6 dpf wildtype and *cdipt* mutant zebrafish were made following the protocol described previously [37]. Briefly, 10–20 embryos from each genotype were dissoci-ated with 3.125 mg/ml collagenase II (Worthington Biochemical Corp., NJ, USA) in $CO_2$-independent media (ThermoFischer Scientific) for 1.5–2 hours. After centrifugation and washes, the pelleted myofibers were resuspended in 1 ml of $CO_2$-independent media and fil-tered through 70 μm and 40 μm filters. The filtered myofiber suspensions were plated on poly-L-lysine-coated coverslips and allowed to settle at room temperature for at least 1 hour. Cover-slips were then fixed immediately with 4% PFA in PBS, rinsed, and stored at 4˚C in PBS until processed for immunostaining.

## Immunofluorescence staining

Immunostaining of myofiber preparations was performed as previously described [37]. Briefly, myofiber preparations were fixed with 4% PFA, for 20 min, permeabilized with PBST (0.3% TritonX in PBS), blocked for 1 hour with PBSTB (5% BSA in PBT) and incubated overnight at 4˚C with primary antibodies. The following primary antibodies were used: mouse anti-DHPR (1:200; DHPRa1A; Abcam), mouse anti-α-Actinin (1:100; Sigma), mouse anti-RyR1 (1:100; 34C; DSHB), rabbit anti-Junctin (1:350; gift from Dulhunty lab), mouse anti-PI(3)P (1:100; Echelon Biosciences Inc.), mouse anti-PI(3,4)P2 (1:100; Echelon Biosciences Inc.), mouse anti-PI(4,5)P2 (1:100; Echelon Biosciences Inc.). Alexa Fluor-conjugated secondary antibodies were used at 1:1000 (Invitrogen). Rhodamine phalloidin (Phalloidin 555) was used to visualize filamentous actin (1:300, Molecular Probes). Preparations were mounted with ProLong Gold with DAPI (Invitrogen). Images were acquired with a Nikon Eclipse Ti laser scanning confocal using NIS Elements software (Nikon Corporation, Tokyo, Japan) and only adjusted for bright-ness and contrast using Adobe Photoshop.

## Live confocal imaging

One-cell stage zebrafish embryos were injected with 10 pg of a cDNA construct containing a fluorescent protein attached to a PIP-binding protein domain [Bodipy-PI (Echelon Biosciences Inc)]; PLC-δ-PH-GFP (Tobias Meyer Lab, Stanford University, CA) using a Picopump (World Precision Instruments). At 1 dpf, injected zebrafish were incubated in 0.2 mM phenylthiourea to prevent pigment formation. To image, zebrafish were screened for fluorescent myofibers on a macroscope (Zeiss Axio Zoom) and mounted in 1.5% low-melt agarose on a 3 cm glass-bottom petri dish. All confocal images were taken with a Nikon Eclipse Ti confocal microscope using a 40x oil-immersion lens.

## Transmission electron microscopy

Zebrafish clutches at 6 dpf were anaesthetised in tricaine and fixed in Karnovsky's fixative overnight at 4˚C. Samples were sent to the Advanced Bioimaging Center (Sickkids Peter Gilgan Centre for Research and Learning, Toronto) where larvae were processed. Briefly, larvae were rinsed in buffer, post-fixed in 1% osmium tetroxide in buffer, dehydrated, and embedded in Quetol-Spurr resin. Following this, 70 nm sections thick were cut with a Leica UC7 ultramicrotome, stained with uranyl acetate and lead citrate, and viewed either with an FEI Tecnai 20 transmission electron microscope (Technai, Oregon, USA) (Bioimaging Facility at The Hospital for Sick Children, Toronto) or with a JEOL JEM 1200EX TEM (JEOL, Massachusetts, USA) (Electron Microscopy Facility at the Laboratory of Pathology, The Hospital for Sick Children, Toronto). Images were obtained using Gatan Digital Micrograph acquisition software or AmtV542, and were manipulated only for brightness and contrast using Adobe Photoshop.

## Immuno-electron microscopy

6 dpf zebrafish embryos were anaesthetised in tricaine and fixed for 2h at room temperature followed by overnight fixation at 4˚C in 4% PFA, 0.1% glutaraldehyde in 0.1M sodium cacodylate buffer with 0.2M sucrose. Samples were rinsed in 0.1M sodium cacodylate buffer and dehydrated in ethanol series (70% ethanol for 1h at 4˚C; 90% ethanol for 1h at 20˚C; 100% ethanol for 1h at -20˚C, twice). Samples were then embedded in 50/50 LR White resin/ethanol for 1h at -20˚C, followed by 70/30 LR White resin/ethanol for 1h at -20˚C and 100% LR White resin for 1h at -20˚C. Samples were then left overnight at -20˚C in 100% LR White resin. Embryos were then placed in capsules filled with LR White resin mixed with benzoin methyl ether (0.1 g in 100 ml LR White), sealed, and placed in the oven for polymerization at 65˚C for at least 72h.

70 nm ultrathin sections were cut with a Leica UC7 ultramicrotome and placed on formvar-coated grids, which were then processed for gold labelled immunostaining. Grids were treated with 0.15M glycine in PBS for 15 minutes, rinsed with PBS, followed by Aurion blocking solution (Aurion, The Netherlands) for 15 minutes. Primary antibodies were diluted in 0.1% BSA-c (Aurion, The Netherlands) at the following concentrations: 1:25 mouse anti-PI $(4,5)P_2$; 1:10 mouse anti-PI$(3,4)P_2$; 1:10 mouse anti-PI$(3)$P (Echelon Biosciences, Inc.). Samples were incubated with primary antibodies for 1h at room temperature. After rinsing the samples with PBS 5 x 5 minutes, these were incubated with 10nm gold-conjugated goat anti-mouse secondary antibody (1:10; Electron Microscope Sciences, Hatfield, PA) for 1h at room temperature. Samples were then rinsed 5 x 5 minutes with PBS, treated with 2% glutaraldehyde (in PBS) for 5 minutes, rinsed 5 x 5 minutes with distilled water and air dried. Gold immunolabelled samples were counter-stained with uranyl acetate and lead citrate and viewed with a JEOL JEM 1200EX TEM (JEOL, Massachusetts, USA) (Electron Microscopy Facility at the Laboratory of Pathology, The Hospital for Sick Children, Toronto). Images were obtained

using AmtV542 software, and were manipulated only for brightness and contrast using Adobe Photoshop.

### Triad size measurement

To determine total triad area (A1+A2+A3), the following features were measured: area of T-tubule (A1), areas of the two terminal cisternae (A2, A3), the distance between the membranes of the two terminal cisternae (D1) and the width of the gap between the membrane of the terminal cisternae and the T-tubule membrane (*). Measurements were done using the open source software Image J. Data and statistical analyses were performed using GraphPad Prism 8 (GraphPad Software Inc., San Diego, CA). Data set for quantification is available on figshare. DOI: https://doi.org/10.6084/m9.figshare.12490589.v1

### Swim test and photoactivation assay

All motor behaviour analysis was performed using Zebrabox software (Viewpoint, France) as previously described [30]. To perform the photoactivation assay, zebrafish were incubated for 5 min at 28.5˚C with optovin 6b8 (ID 5705191l; ChemBridge), an optovin analog [38]. Optovin is a reversible TRPA1 ligand that elicits motor excitation following exposure to light. After incubation, the Zebrabox platform monitored larvae for 20 second cycles over 10 minutes. Parameters were set to capture 5 seconds of exposure to white light to elicit ambulatory movement, followed by 15 seconds of recovery behaviour in the dark. This was repeated 30 times, to get a total experiment time of 10 minutes. The average speed traveled during the 20 second cycle was used to compare groups (i.e., *cdipt* mutants vs. WT siblings). To perform the spontaneous swim assay, zebrafish in system water were followed for 1 hour. Data were analyzed using statistics software (GraphPad Prism). Data set for quantification is available on figshare. DOI: https://doi.org/10.6084/m9.figshare.12490589.v1

### Morpholino studies

For knockdown of maternal *cdipt*, the following ATG-targeting MO was designed: 5′-CCG AGAGTTTCTTTCTTTGGACGGA-′3 (GeneTools LLC). An MO designed to a random sequence (5′-CCTCTTACCTCAGTTACAATTTATA-3′) with no homology by Basic Local Alignment Search Tool (BLAST) analysis in the zebrafish genome was used as a control (GeneTools LLC). Fertilized eggs were collected after timed matings of adult zebrafish and injected at the 1-cell stage using a Picopump (World Precision Instruments). Embryos were injected with concentrations ranging from 0.15–0.5 mM in a volume of 1 nl.

## Results

### Developing a new *cdipt* mutant zebrafish line

Exon 3 of the *cdipt* gene was targeted using the CRISPR/Cas9 system (Fig 1B). A 10 bp deletion allele, hereafter referred to as *cdipt* mutant when present in homozygosity, was identified after Sanger sequencing (Fig 1C). Due to lack of commercially available antibodies against zebrafish CDIPT, we performed real-time PCR (qPCR) on total RNA from whole embryos to confirm that *cdipt* transcript is reduced by this mutation. There was a significant difference in *cdipt* mRNA levels between wildtype (WT) and mutants at both 3 dpf and 6 dpf (Fig 1D) (P < 0.05, n = 30). This suggests that mutant *cdipt* mRNA transcripts are being directed to the nonsense-mediated decay pathway, and is consistent with this mutation being a loss of expression and function allele.

## *cdipt* zebrafish exhibit morphological and gastrointestinal system abnormalities

Homozygous *cdipt* mutant fish appeared phenotypically normal until 5 dpf, when gastrointestinal system abnormalities are visible with bright-field microscopy. The mutant phenotype is fully penetrant at 6 dpf and includes a dark, globular liver and small intestine, partial deterioration of the ventral fin (folds, incisions, missing areas), tissue degradation around the cloaca, and abnormal jaw structure (Fig 2A and S1 Fig). The gastrointestinal features are reminiscent of the mutant *cdipt*<sup>hi559/hi559</sup> phenotype, and have already been well characterized [35]. Of note, we validated our *cdipt* mutant by crossing heterozygous fish with *cdipt*<sup>hi559/+</sup> fish (obtained from ZIRC). The resulting compound heterozygous offspring manifested a liver phenotype identical in appearance to both *cdipt*<sup>hi559/hi559</sup> and our newly generated *cdipt* mutant.

## *cdipt* zebrafish have generally normal muscle structure

Gross morphology of *cdipt* zebrafish muscle was investigated using birefringence. Birefringence uses polarized light to assess muscle integrity. Organized skeletal muscle will appear bright amidst a dark background when visualized between two polarized light filters, whereas disorganized muscle exhibits degenerative dark patches and an overall decrease in brightness

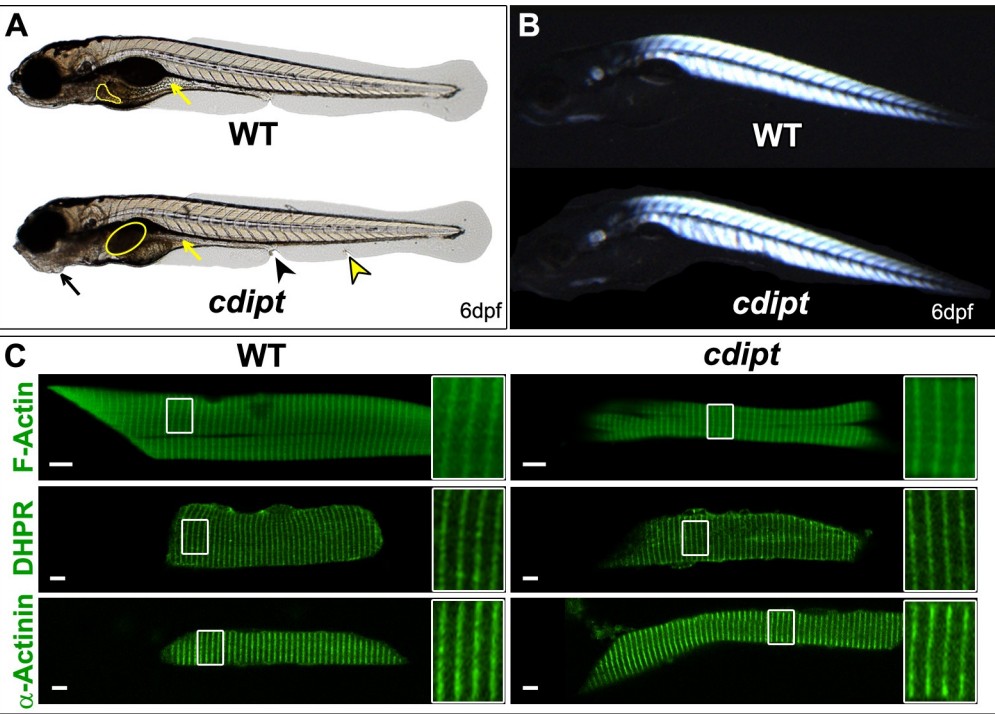

**Fig 2. Characterization of the *cdipt* mutant phenotype at 6dpf. A)** *cdipt* mutant zebrafish exhibit a gastrointestinal phenotype with a dark, globular and oversized liver (yellow outline) and a small intestine (yellow arrows), abnormal jaw structure (black arrows), tissue degradation around the cloaca (black arrowhead), and defective ventral fin (yellow arrowhead). **B)** Representative image of *cdipt* mutants at 6dpf, showing normal birefringence pattern indistinguishable from WT siblings, indicative of normal sarcomere organization. **C)** Confocal micrographs showing localization by indirect immunofluorescence of actin (upper panels), DHPR (middle panels) and α-actinin (bottom panels) in the skeletal myofibers. There is no noticeable difference in the localization of these proteins between WT (left column) and *cdipt* mutant (right column). Insets represent high magnification of areas surrounded by white rectangles. Scale bars = 5 μm.

in some myotomes. Based on birefringence analysis, *cdipt* zebrafish have normal muscle integrity and sarcomere organization at all ages examined (Fig 2B).

We next studied the localization of several sarcomeric proteins, such as actin, myosin (contractile proteins), dihydropyridine receptor (DHPR), ryanodine receptor type 1 (RyR1) and junctin (markers for triads), laminin and dystrophin (markers for myotendinous junctions), and α-actinin (a Z-line marker). Immunostaining with antibodies against these proteins on myofibers isolated from 6dpf zebrafish showed no differences in localization between WT and *cdipt* mutants (Fig 2C and S2 Fig), indicating no qualitative defects in the formation and organization of key muscle structures.

We next studied the ultrastructure of muscle, given that abnormal triad formation is a hallmark of many PIP-related myopathies and may not be appreciated by light microscopy. *cdipt* larvae and their WT siblings were thus processed at 6 dpf for transmission electron microscopy. Electron micrographs revealed no major abnormalities in triad structure in *cdipt* mutants (Fig 3A and 3B). To better characterize the triads we measured the area of T-tubules (A1), the areas of the two terminal cisternae (A2, A3), total triad area (A1+A2+A3), the distance between the membranes of the two terminal cisternae of the sarcoplasmic reticulum (D1), and the width of the gap between the T-tubule and each of the two terminal cisternae (*), where the junctional feet corresponding to the ryanodine receptor-dihydropyridine receptor complex are found [4] (Fig 3C). There was no significant difference in the total area of the triad (A1+A2+A3) (Fig 3D). However, the distance between terminal cisternae at maximum distance (D1) and the gap width (*) were quantitatively and significantly smaller in *cdipt* mutants (Fig 3D).

## *cdipt* zebrafish have abnormal motor behaviour as compared to wildtype siblings

Given the subtle but significant change observed in the appearance of the triad, we wanted to determine if there was any alteration in muscle function. To assess muscle function, we performed a spontaneous swim test assay and a routine photoactivation movement assay previously utilized by our lab [30]. The latter involved incubating zebrafish larvae with a molecule called optovin 6b8, which when exposed to white light, will activate zebrafish muscle through a reflex arc. If muscle function is impaired, mutants will have reduced movement when compared to WT. We did thirty rounds of 20 second-activation periods to assess both the speed of movement and muscle fatigue. The average speed of movement was significantly lower in *cdipt* mutant zebrafish both in the spontaneous swim test (Fig 4A and 4B) and in their response to optovin (Fig 4C and 4D). In addition, *cdipt* mutants spent less time moving (S3A Fig) and covered shorter distances than their WT siblings (S3B Fig). The rate of fatigue, however, was similar in the *cdipt* mutant and WT zebrafish, with both groups reaching a plateau at the same rate (Fig 4D).

## *cdipt* zebrafish do not show changes in the localization of PIPs

Given that CDIPT is the rate-limiting enzyme for PI synthesis (the precursor for all PIPs), we looked at the expression and localization of several species of PIPs in myofibers. We specifically investigated PI3P, PI(3,4)P2 and PI(4,5)P2 localization. PI(4,5)P2 and PI(3,4)P2 are found mostly at the plasma membrane, whereas PI3P is mostly found on endosomes [13]. Immunofluorescence staining with anti-PIP antibodies revealed no significant differences in the localization of PI(4,5)P2, PI3P, PI(3,4)P2 (Fig 5). To further investigate PI(4,5)P2 localization in *cdipt* larvae *in vivo*, a fluorescent marker for PI(4,5)P2 (PLCδPH-GFP) was injected into 1-cell stage *cdipt* embryos. At 6 dpf, PI(4,5)P2 appeared to properly localize to the plasma

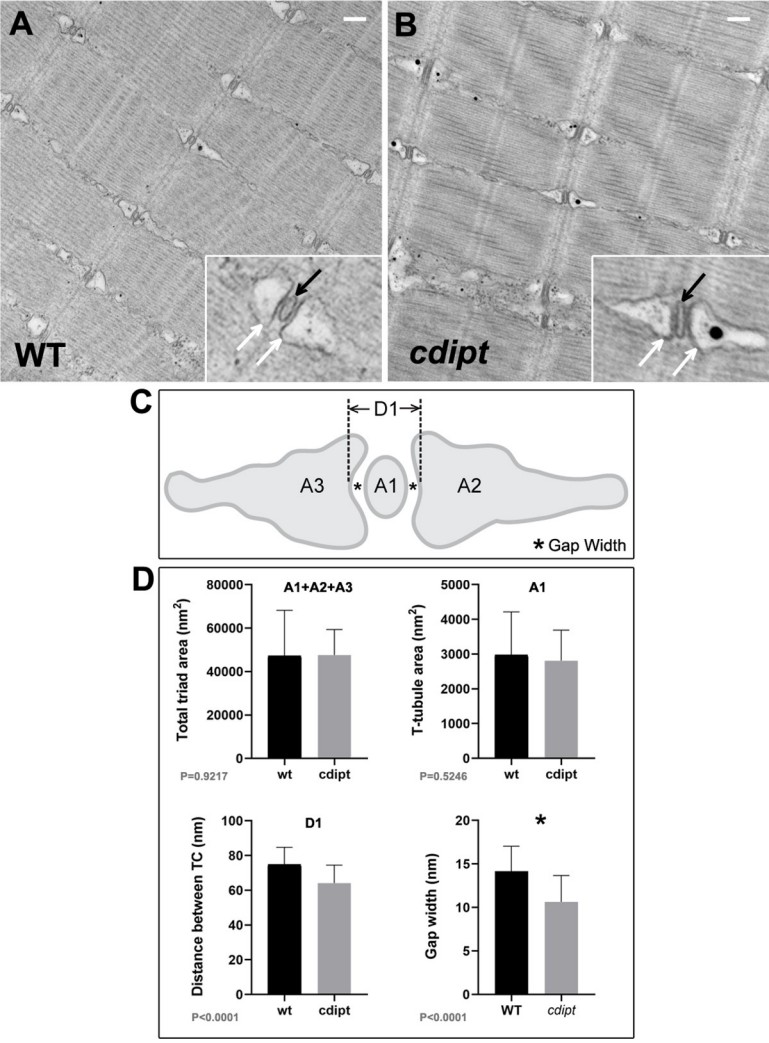

**Fig 3. Skeletal muscle ultrastructure. A-B)** Transmission electron micrographs show normal skeletal muscle ultrastructure in *cdipt* larvae. T-tubules (insets; black arrow) are apposed by terminal cisternae of sarcoplasmic reticulum (insets; white arrow). **C)** Diagram illustrating triad structure and features used for measurements: A1 = T-tubule area; A2 and A3 = terminal cisternae (TC) areas; D1 = maximum distance between TCs; * = gap width (distance between TC membrane and T-tubule membrane). **D)** There is no significant difference in the triad area between WT and *cdipt* mutant larvae (A1+A2+A3 graph) (n = 36, p = 0.9217). The T-tubule area (A1 graph) is qualitatively slightly smaller in the *cdipt* mutant than in WT (n = 36; P = 0.5246), whereas the distance between cisternae at maximum distance (D1 graph) (n = 36, p < 0.0001) and the gap width (* graph) (n = 44, p < 0.0001) are significantly smaller in *cdipt* mutants than in WT. Scale bars = 200 nm.

membrane (Fig 5A and 5A'). These results were further supported by immunoelectron microscopy studies. Nanogold-labelled antibodies against PI3P, PI(3,4)P2 and PI(4,5)P2 localized at the triad and its vicinity and showed similar localization pattern in WT and *cdipt* mutant embryos (Fig 6).

## Maternal *cdipt* mRNA and/or PI are sufficient for normal muscle development

Given the importance of CDIPT in generating a precursor to all PIP species, it is surprising that there is no developmental phenotype in skeletal muscle. However, a previously published

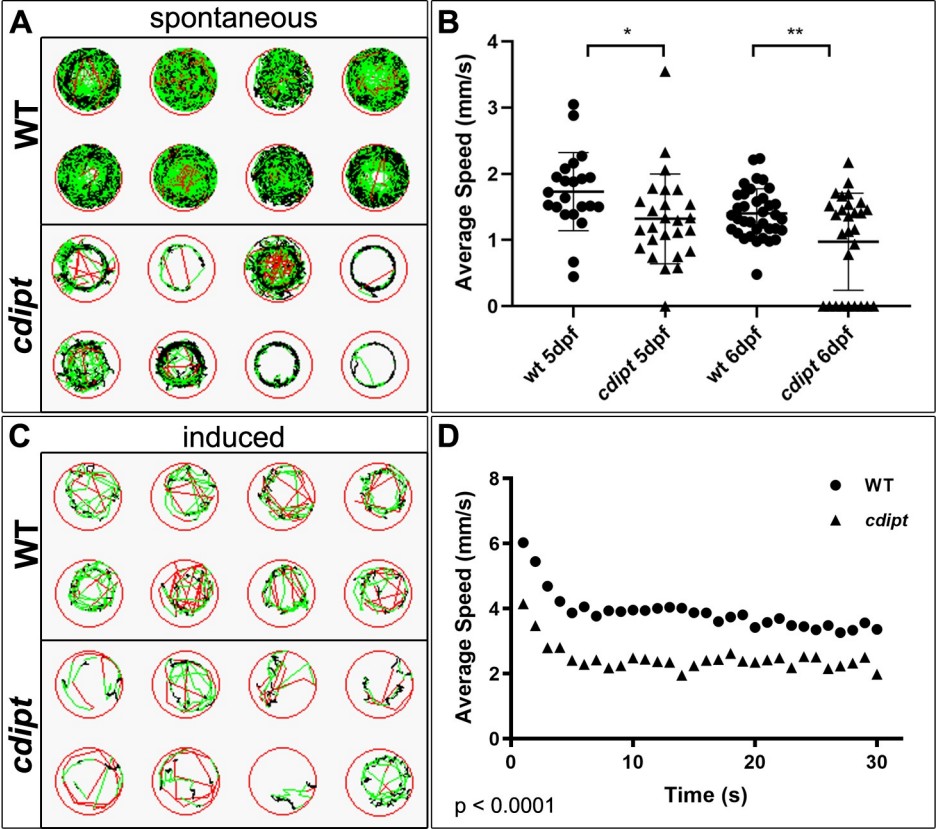

**Fig 4.** *cdipt* **mutants have significantly impaired motor function compared to their wildtype siblings. A)** Spontaneous swim movement was assessed by tracking 5-days or 6-days old zebrafish larvae over 1 hour. Representative examples of tracking plots of individual larvae movement. Black represents slow movement (<5 mm/s), green represents average speed (5–20 mm/s), and red represents fast movement (>20 mm/s). **B)** The *cdipt* mutant larvae are significantly slower than their WT siblings, both at 5dpf (WT n = 22, *cdipt* n = 26, p = 0.0318) and 6dpf (WT n = 36, *cdipt* n = 28, p = 0.0036). **C)** Involuntary motor function was assessed using an optovin-stimulated movement assay in response to pulses of light. Representative examples of tracking plots of individual larvae showing movement over 20 seconds, involving 5 seconds of white light exposure followed by 15 seconds of darkness. **D)** There is a significant difference between the average speed travelled by WT zebrafish compared to *cdipt* mutant zebrafish (n = 18 and 14, respectively; p < 0.0001, Student's *t* test, 2-tailed). WT and *cdipt* mutant zebrafish plateau at the same rate (n = 7, p = 0.3487).

lipidomic analysis of the early zebrafish yolk found that PI is already present at 0 hpf [39] and previous results on *cdipt*^hi559/hi559^ zebrafish [35] and our qPCR results (see Fig 1D) show there is maternal *cdipt* mRNA expression in early stages of the zebrafish embryo before zygotic gene expression is turned on.

To prevent production of CDIPT protein from maternal mRNA, we injected zebrafish embryos at one-cell stage with a translation blocking morpholino (ATG-MO). Injection of *cdipt* ATG-MO at both 0.3 mM and 0.5 mM caused increased levels of embryo death (S4A Fig), suggesting that blocking of maternal *cdipt* mRNA translation is broadly detrimental for embryogenesis. However, some morphants did survive beyond 3 dpf. In these morphants, the skeletal muscle development was not obviously affected, as assessed by general inspection and by birefringence (S4B Fig).

To investigate whether maternally deposited PI in the yolk can be delivered to developing skeletal muscle, we injected BODIPY-labelled PI into yolk at the one-cell stage. Fluorescence was tracked with a confocal microscope over several days. By 1 dpf, the fluorescent probes

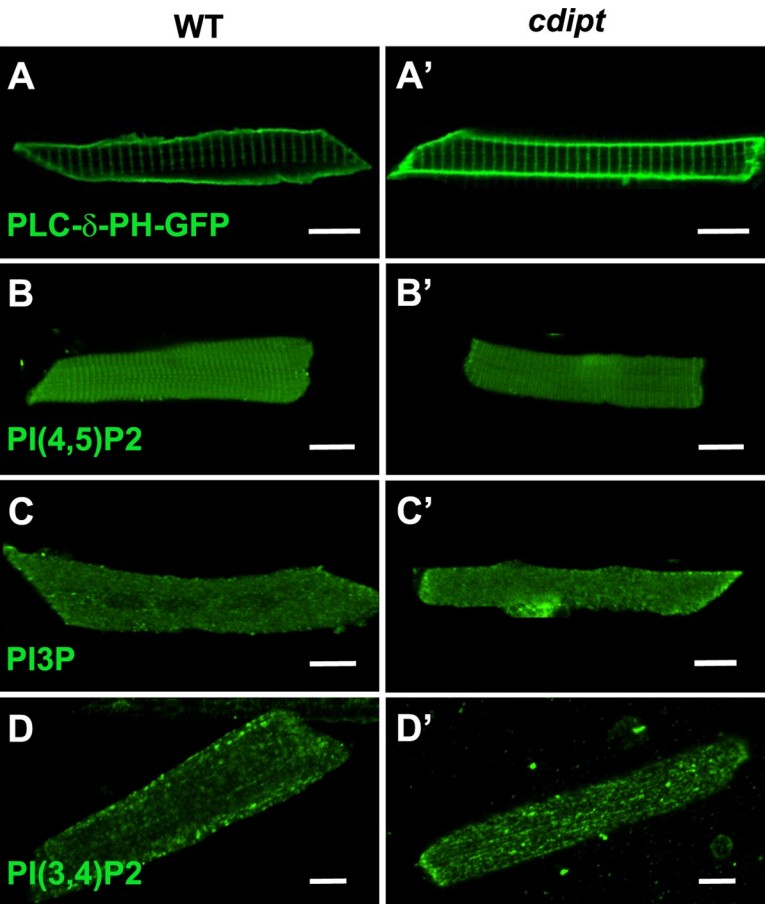

**Fig 5. Localization of PIPs by immunofluorescence in wildtype and *cdipt* mutant zebrafish.** Confocal micrographs showing localization of PIPs is not affected in early larval development of *cdipt* mutants. **(A, A')** visualization of skeletal muscle from live embryos injected with PLCδPH-GFP, a marker for $PI(4,5)P_2$. There was no obvious difference in expression between wild type (WT) and *cdipt* mutant embryos. **(B-D, B'-D')** Immunostaining with PIP antibodies of myofibers isolated from WT and *cdipt* mutants. Localization of $PI(4,5)P_2$ **(B, B')**, PI3P **(C, C')** and $PI(3,4)P_2$ **(D, D')** is similar in wildtype and *cdipt* zebrafish. Scale bars = 10μm.

appeared in the skeletal muscle compartment (S5 Fig). Fluorescence was not detectable at 2 days post-injection and later. These results suggest that PI present in the yolk at early stages can be delivered to skeletal muscle.

Taken together, our data suggest that the presence of maternally deposited mRNA in the cell and/or PI in the yolk fulfill the early developmental requirements for CDIPT and for PI, which is consistent with the lack of a phenotype until the yolk is depleted at 5 dpf. This also suggests that once PI and its PIP derivatives are generated, they likely persist as a stable pool in skeletal muscle.

## Discussion

To investigate the role of PIPs in muscle development, we developed and characterized a new *cdipt* mutant zebrafish. This mutant showed defects in fin morphology and aberrant swimming behaviour, in addition to the gastrointestinal defects previously reported in another *cdipt* mutant (*cdipt^{hi559/hi559}*) [35].

The purpose of generating this model was to determine how the potential loss of all seven PIP species would affect muscle development. We expected that loss of CDIPT and the

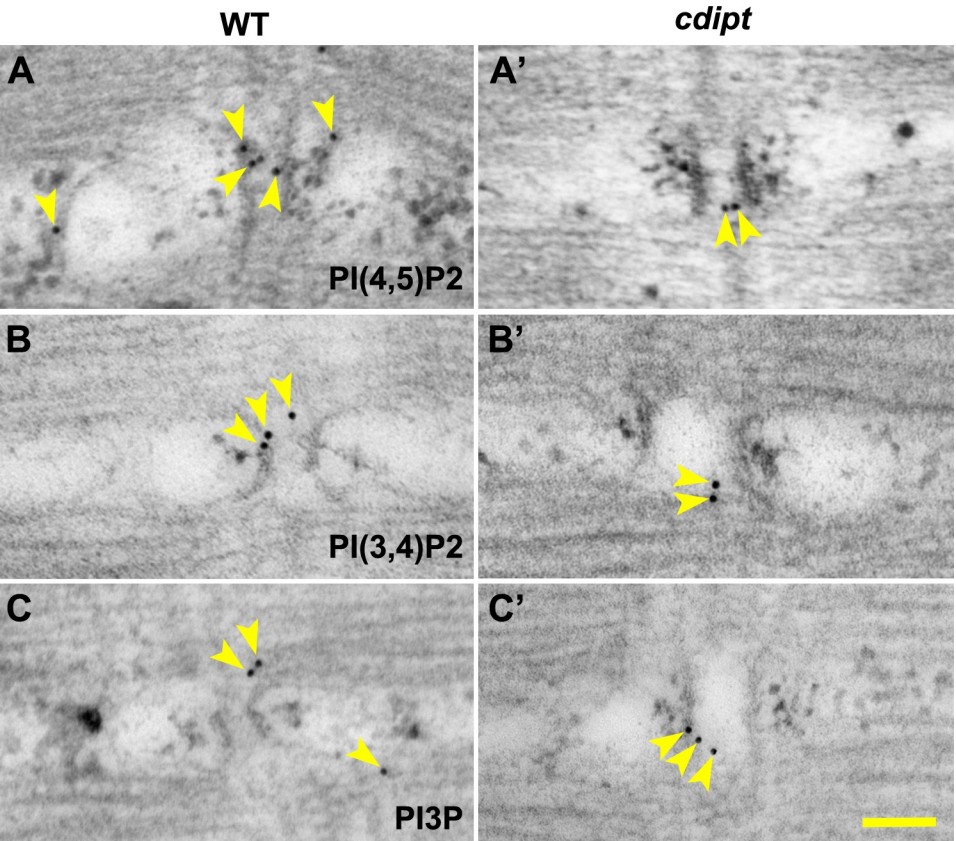

**Fig 6. Localization of PIPs by immunoelectron microscopy in wildtype and *cdipt* mutant zebrafish.** Transmission immunoelectron micrographs showing localization of nanogold-labelled antibodies against **(A, A')** PI(4,5)P$_2$, **(B, B')** PI(3,4)P$_2$ and PI3P **(C, C')** (yellow arrowheads) at the skeletal muscle triad. There is no difference in localization of these antibodies between WT and *cdipt* mutant embryos. Scale bar = 100nm.

subsequent depletion of PI would have severe effects on skeletal muscle; however, mutant *cdipt* larvae showed only minimal abnormalities in muscle structure and overall muscle function. We hypothesize that this modest phenotype is the result of PI deposited in the muscle during embryogenesis that then provides sufficient substrate for generation and maintenance of PIP species at subsequent developmental stages.

Given the importance of CDIPT in generating a precursor to all PIP species, it is surprising that there is no significant adverse phenotype in muscle. The lack of widespread abnormalities may be because *cdipt* mRNA and PIs are maternally deposited into zebrafish yolk. Depletion of maternal mRNA using a translation-blocking morpholino resulted in increased mortality in the zebrafish, consistent with a role for CDIPT and de novo PI synthesis in embryogenesis. However, the impact of the translation-blocking morpholino did not seem to affect skeletal muscle development in the surviving injected embryos, suggesting that maternally deposited CDIPT does not play a role in myogenesis. There is likely also an important contribution to total embryo PI from maternal deposition of this precursor lipid in the yolk. In order to remove this potential confounder to the assessment of CDIPT function in skeletal muscle, we would need to prevent the deposition of PI into the yolk, or develop a method of depleting yolk PI without disrupting other essential nutrients contained within the yolk. Currently, there are no technologies that would allow us to complete either of those experiments. We attempted

to use direct lipase injection into the yolk in order to deplete it, but this resulted in embryonic lethality prior to myogenesis.

The one part of the muscle where we did observe abnormalities was the triad. We showed by immunofluorescence that several species of PIPs localize to the sarcomere. Moreover, our novel immunoelectron microscopy studies showed these molecules localize to the triad. To our knowledge, this is the first report on ultrastructural localization of PI3P, PI(3,4)P2 and PI (4,5)P2 at the triad in a vertebrate model, as previous studies focused on culture cell lines [40– 44]. Our data add to the growing evidence showing the importance of PIP metabolism in the development and maintenance of this key muscle substructure. Work with mammalian myocytes in culture has implicated PIP2 (via its binding to BIN1) in the formation of the T-tubule [20, 45]. While we did not see an overall decrease in PIP2 levels, nor in its localization, it is tempting to speculate that the loss of CDIPT impacted PIP synthesis sufficiently to result in mild but critical reduction in PIP2 levels that were enough to alter triad formation. Specifically, this alteration might affect binding to and localization of BIN1 to the T-tubule microdomains, the result of which would impair triad formation and/or maintenance and result in altered triad function and muscle weakness.

Alternatively, the motility impairment we see in the *cdipt* mutants might be due to defects in the organization and function of the neuromuscular junctions (NMJ). Phosphoinositides have been shown to regulate NMJ growth and morphology [46] and synaptic vesicle recycling in fruit flies [47, 48]. In addition, NMJs are disrupted in preclinical models of centronuclear myopathy due to mutations in MTM1 or DNM2 [49, 50]. We did explore this possibility by analyzing fatigue, a functional measure of NMJ signalling. We did not detect any signs of fatigable muscle weakness in our mutants, which would argue against a contribution to the motility impairment by disrupted NMJs. However, future studies will be required to fully explore this alternative mechanism and its potential relevance to the *cdipt* mutant motor phenotype.

Of note, phenotypic abnormalities in *cdipt* mutants do not appear until 5 dpf, shortly after the yolk has been depleted. The most prominent of these is the digestive system phenotype, likely reflecting a requirement for *de novo* PIP synthesis in this organ system, since pools of PI are locally made and used almost immediately after synthesis [51]. However, because skeletal muscle has no phenotype at 5 dpf and previous data has shown that *cdipt* mRNA is not present in skeletal muscle after 5 dpf [51], it is possible that maturing skeletal muscle does not require *de novo* PIP synthesis. Instead, perhaps PIPs are maintained in pools that can fluctuate between the different species when needed. Alternatively, a requirement for PIP synthesis in muscle may not manifest in the window of time after yolk depletion and before mutant death, but may develop as the muscle continues to grow and mature. Muscle specific targeting of *cdipt* would be helpful in the future to distinguish between these possibilities.

Of note, the most obvious phenotypes in the *cdipt* mutants are in the liver, gastrointestinal system, and the fin. Interestingly, these phenotypes of the *cdipt* mutants are also visible in *mtm1* mutant zebrafish [30]. The fact that two mutated PIP-related genes cause similar defects suggest that PIP metabolism must be tightly regulated in these tissues in zebrafish development, and that there is an increased requirement for de novo synthesis and homeostatic balance.

## Supporting information

**S1 Fig. Phenotypic variations in *cdipt* mutants. A-B)** Examples of fin degeneration (yellow arrowheads), oversized liver (yellow outline), and abnormal jaw structure (black arrows) in *cdipt* mutant zebrafish. **C)** Many *cdipt* mutants have partially folded ventral fin. (TIF)

**S2 Fig. Localization of triad-associated proteins in the muscle.** Confocal micrographs showing localization by indirect immunofluorescence of RyR1 (top panels) and Junctin (bottom panels) in skeletal myofibers. There is no noticeable difference in localization of these proteins in WT (left panels) and *cdipt* mutant (right panels). Scale bars = 10μm.
(TIF)

**S3 Fig. *cdipt* mutants have impaired motor function. A)** *Cdipt* mutant zebrafish spend significantly less time swimming compared to their WT siblings, both at 5dpf (WT n = 22, *cdipt* n = 26, p = 0.0359) and 6dpf (WT n = 36, *cdipt* n = 28, p = 0.0209). **B)** *Cdipt* mutant zebrafish travel significantly shorter distances compared to their WT siblings at 5dpf (WT n = 22, *cdipt* n = 26, p = 0.0121) whereas at 6dpf the travelled distances are not significantly different (WT n = 36, *cdipt* n = 28, p = 0.1015).
(TIF)

**S4 Fig. Blocking maternal *cdipt* mRNA translation is detrimental for embryogenesis. A)** Embryos injected with ATG-MO (n = 115) show significantly higher mortality rates than those injected with Ctrl-MO (n = 117). **B)** Surviving ATG-MO-injected *cdipt* embryos have a normal birefringence pattern indistinguishable from their WT siblings.
(TIF)

**S5 Fig. Maternally deposited PI in the yolk is transported to the muscle.** Zebrafish larvae at 1 dpf after injection of BODIPY-PI into yolk at the 1-cell stage (arrows indicate accumulation of fluorescently-labeled PI in the muscle).
(TIF)

## Acknowledgments

The authors gratefully thank Jonathan Volpatti, Yukari Endo and Mo Zhao (Dowling Lab) for useful discussions and insightful suggestions; Evangelina Aristegui for technical support; Scott Knox, Alejandro Salazar, and Elyjah Schimmens for zebrafish care (SickKids Zebrafish Facility); Paul Paroutis and Kimberley Lau for technical support (SickKids Imaging Facility); and Doug Holmyard, Ali Darbandi and William Martin (SickKids Nanoscale Biomedical Imaging Facility) for help with TEM and immunoEM sample preparation. The authors also thank the following people for generously providing reagents: Angela Dulhunty (John Curtin School of Medical Research, Australian National University, Canberra, Australia) for anti-Junctin antibody; Tobias Meyer Lab (Stanford University, CA) for the PLC-δPH-GFP construct; Sergio Grinstein and Julie Brill (The Hospital for Sick Children, Toronto, Canada) for PIP constructs and for helpful discussion. The authors gratefully acknowledge Zebrafish International Resource Center for providing the *cdipt*^hi559^ fish line.

## Author Contributions

**Conceptualization:** Lindsay Smith, James J. Dowling.

**Data curation:** Lindsay Smith, Lacramioara Fabian.

**Formal analysis:** Lindsay Smith, Lacramioara Fabian, Ramil R. Noche.

**Funding acquisition:** James J. Dowling.

**Investigation:** Lindsay Smith, Lacramioara Fabian, Almundher Al-Maawali, Ramil R. Noche.

**Methodology:** Lindsay Smith.

**Project administration:** Lindsay Smith, Lacramioara Fabian, James J. Dowling.

**Supervision:** James J. Dowling.

**Validation:** Lindsay Smith, Lacramioara Fabian.

**Visualization:** Lindsay Smith, Lacramioara Fabian.

**Writing – original draft:** Lindsay Smith.

**Writing – review & editing:** Lindsay Smith, Lacramioara Fabian, Almundher Al-Maawali, Ramil R. Noche, James J. Dowling.

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
