## [Decision Letter · Decision Letter 0]

16 Apr 2020

PONE-D-20-07805

De Novo Phosphoinositide Synthesis in Zebrafish Is Required for Triad Formation but Not Essential for Myogenesis

PLOS ONE

Dear Dr. James Dowling,

Thank you for submitting your manuscript to PLOS ONE. After careful consideration, we feel that it has merit but does not fully meet PLOS ONE’s publication criteria as it currently stands. Therefore, we invite you to submit a revised version of the manuscript that addresses the points raised during the review process.

We would appreciate receiving your revised manuscript by May 31 2020 11:59PM. To enhance the reproducibility of your results, we recommend that if applicable you deposit your laboratory protocols in protocols.io, where a protocol can be assigned its own identifier (DOI) such that it can be cited independently in the future. For instructions see: http://journals.plos.org/plosone/s/submission-guidelines#loc-laboratory-protocols

We look forward to receiving your revised manuscript.

Kind regards,

Seungbok Lee, Ph.D.

Academic Editor

PLOS ONE

Journal Requirements:

Reviewers' comments:

Reviewer's Responses to Questions

**Comments to the Author**

1. Is the manuscript technically sound, and do the data support the conclusions?

Reviewer #1: Yes

Reviewer #2: Yes

2. Has the statistical analysis been performed appropriately and rigorously? 

Reviewer #1: Yes

Reviewer #2: Yes

3. Have the authors made all data underlying the findings in their manuscript fully available?

Reviewer #1: Yes

Reviewer #2: No

4. Is the manuscript presented in an intelligible fashion and written in standard English?

Reviewer #1: Yes

Reviewer #2: Yes

5. Review Comments to the Author

Reviewer #1: In the present paper, the authors performed the loss of function analysis of CDP-diacylglycerol-inositol 3-phosphatidyltransferase (CDIPT) in zebrafish skeletal muscle development. They generated CDIPT mutant zebrafish, analyzed structures of skeletal muscles, motor activity and ultrastructural localization of phosphoinositides (PIPs) derivatives in mutant larvae, and found that the CDIPT mutation causes abnormalities of triad formation and motor activity with normal embryonic muscle development. These results suggest the role of CDIPT in triad formation and motor function. However, in order to strengthen the manuscript, following comments need to be clarified prior to publication.

Comments

1) Line 150: In the section of “Material and methods,” I hope the authors to add concise protocol of the “Skeletal myofiber preparations” for readers.

2) In the panel D of Figure 3, the graph format of gap width is different with other graphs. I recommend the authors to unify format of the graphs in D.

3) Line 309-337: In the “Results” section, the authors showed abnormal triad formation in the mutant, using electron microscopy. Especially, they found that gap width was significantly decreased and the T-tubule area (A1) is slightly smaller in the mutant. I wonder how these changes induce abnormal motor activity in the mutant. It is required to describe possible explanations for that, even if it could be hypothetical mechanisms.

4) Line 352-353, 369-370: In the “Results” and “Figure 4 legend”, the authors explained the rate of fatigue in the wildtype and mutant. To explain the rate of fatigue clearly, the sentence in the figure legend (Line 369-370) needs to be positioned in the “Result” section.

5) Line 415-422: In the “Results” section, the authors performed knockdown experiment to block maternal cdipt mRNA, using a translation blocking morpholino. They mentioned that cdipt MO caused level of embryonic death in the manuscript. However, the survival rate seems to be high until 3dpf in the Supplementary Figure 4. The authors need to check the inconsistency of survival rate of the cdipt morphants.

Reviewer #2: This manuscript shows a new phenotype in CDITP mutant zebrafish. The analysis is appropriate and the fact that there are significant motor defects is very interesting. These defects are attributed to an exceedingly subtle defect in the triad that apparently requires de novo PI synthesis. Unfortunately, the authors did not analyze the neuromuscular junction which also contributes to movement. It might be reasonable for the authors to recognize this in the discussion.

Major

The authors state that CDITP is the only protein currently predicted to generate PI and reference a paper over 5 years old. Given improvements to the genome sequence, it would be helpful if the authors state whether they undertook a careful investigation of the genome to identify possible related genes – the rationale that in addition to maternally deposited CDITP and the fairly stable pool it is possible that there is another gene. It would also be helpful if they explained why they made a new mutation.

Minor

Abstract: This is written as a specialized paper for readers who know what the triad is – it would be handy for the authors to include some of the verbiage in their introduction regarding the hypothesized role of the triad in muscle weakness in myopathies.

Line 63: subject is dysregulation verb should be has not have

Lines 79-85 – presumably more recent papers regarding zebrafish as model for muscle development could also be referenced, and other references should be included for the subsequent statements.

Line 320: the qualitatively smaller is a huge stretch with the data shown, I would suggest taking that out.

6. PLOS authors have the option to publish the peer review history of their article (what does this mean?). If published, this will include your full peer review and any attached files.

Reviewer #1: No

Reviewer #2: No

---

## [Author Response · Author response to Decision Letter 0]

17 Jun 2020

Please see uploaded Response-to-Reviewers letter

---

## [Decision Letter · Decision Letter 1]

3 Aug 2020

De Novo Phosphoinositide Synthesis in Zebrafish Is Required for Triad Formation but Not Essential for Myogenesis

PONE-D-20-07805R1

Dear Dr. Dowling,

We’re pleased to inform you that your manuscript has been judged scientifically suitable for publication and will be formally accepted for publication once it meets all outstanding technical requirements.

Kind regards,

Seungbok Lee, Ph.D.

Academic Editor

PLOS ONE

Additional Editor Comments (optional):

Reviewers' comments:

Reviewer's Responses to Questions

**Comments to the Author**

1. If the authors have adequately addressed your comments raised in a previous round of review and you feel that this manuscript is now acceptable for publication, you may indicate that here to bypass the “Comments to the Author” section, enter your conflict of interest statement in the “Confidential to Editor” section, and submit your "Accept" recommendation.

Reviewer #1: All comments have been addressed

2. Is the manuscript technically sound, and do the data support the conclusions?

Reviewer #1: Yes

3. Has the statistical analysis been performed appropriately and rigorously? 

Reviewer #1: Yes

4. Have the authors made all data underlying the findings in their manuscript fully available?

Reviewer #1: Yes

5. Is the manuscript presented in an intelligible fashion and written in standard English?

Reviewer #1: Yes

6. Review Comments to the Author

Reviewer #1: I think the authors responded appropriately to reviewer comments and that the current version of the manuscript is nicely done. I believe this work will make an important contribution to the field.

7. PLOS authors have the option to publish the peer review history of their article (what does this mean?). If published, this will include your full peer review and any attached files.

Reviewer #1: No

---

## [Editor Report · Acceptance letter]

6 Aug 2020

PONE-D-20-07805R1 

De Novo Phosphoinositide Synthesis in Zebrafish Is Required for Triad Formation but Not Essential for Myogenesis 

Dear Dr. Dowling:

I'm pleased to inform you that your manuscript has been deemed suitable for publication in PLOS ONE. Congratulations! Your manuscript is now with our production department. 

Kind regards, 

on behalf of

Dr. Seungbok Lee 

Academic Editor

PLOS ONE